# Enhanced Visible-Light-Driven Photocatalysis of Ag/Ag_2_O/ZnO Nanocomposite Heterostructures

**DOI:** 10.3390/nano12152528

**Published:** 2022-07-23

**Authors:** Chadrasekhar Loka, Kee-Sun Lee

**Affiliations:** Department of Advanced Materials Engineering & Smart Natural Space Research Center, Kongju National University, Cheonan 31080, Korea

**Keywords:** surface plasmon resonance, organic dye, silver, hydrothermal, photoluminescence, optical properties

## Abstract

Visible-light-driven photocatalysis is one promising and efficient approach for decontaminating pollutants. Herein, we report the combination of localized surface plasmon resonance (LSPR) and p-n heterojunction structure Ag-Ag_2_O-ZnO nanocomposite synthesized by a hydrothermal process for the suppression of photogenerated electron-hole pair recombination rates, the extension of the absorption edge to the visible region, and the enhancement of photocatalytic efficiency. The prepared nanocomposites were investigated by standard analytical techniques and the results revealed that the synthesized powders were comprised of Ag, Ag_2_O, and ZnO phases. Photocatalytic activity of the photocatalyst tested for methylene blue, methyl orange, and rhodamine B showed the highest photocatalytic degradation efficiency: 97.3%, 91.1%, and 94.8% within 60 min under visible-light irradiation. The average lifetime of the photogenerated charge carriers was increased twofold in the Ag-Ag_2_O-ZnO photocatalyst (~10 ns) compared to the pure ZnO (~5.2 ns). The enhanced photocatalytic activity resulted from a decrease of the charge carrier recombination rate as inferred from the steady-state and time-resolved photoluminescence investigations, and the increased photoabsorption ability. The Ag-Ag_2_O-ZnO photocatalyst was stable over five repeated cyclic photodegradation tests without showing any significant changes in performance. Additionally, the structure indicated a potential for application in environmental remediation. The present study showcases the robust design of highly efficient and reusable visible-light-active photocatalysts via the combination of p-n heterojunction and LSPR phenomena.

## 1. Introduction

Environmental pollution has become a major problem due to the world’s ever-growing population, as well as the huge discharge of hazardous industrial waste, such as organic dyes, food waste, and other substances, into the environment. Particularly, the substantial amount of polluted water released into the environment by numerous industries such as textiles, paper printing, pharmaceuticals, leather, and cosmetics have become an overwhelming problem for modern public health. Therefore, the decontamination of hazardous effluents is imperative due to their harmful effect on human, animal, and water ecology. Numerous semiconductor metal-oxide photocatalysts such as TiO_2_, ZnO, SnO_2_, Fe_2_O_3_, and BiVO_4,_ have been considered for environmental remediation, however, their practical usage is severely limited because they are ultraviolet-light active, and ultraviolet light covers approximately 4% of the solar spectrum. In this context, most research has been devoted to the exploitation of visible light, which constitutes about 45% of the solar spectrum. ZnO has been proven to be a dynamic and safe photocatalyst, widely used for wastewater treatment, due to its high light absorption coefficient, high redox potential, high quantum efficiency, low cost, and nontoxicity. Additionally, ZnO works under ultraviolet light [1]. Moreover, the practical application of bare photocatalysts is limited because of their poor recyclability caused by agglomeration or photodegradation. Various strategies have been reported to exploit ZnO for visible light photocatalysis, including modification of the nanostructure, doping of metals or nonmetals, construction of heterojunction, upconversion of nanoparticles, or coupling of carbon materials [2,3,4]. Among them, heterojunction structures comprised of two or more semiconductors, typified as g-C_3_N_4_/ZnO [5], TiO_2_/ZnO [6], Fe_3_O_4_/ZnO [7], Co_3_O_4_/ZnO [8], CuO/ZnO [9], have gained lots of attention because of their synergetic effects on photocatalytic performance, including enhanced interfacial charge transfer, prolonged carrier lifetime, and higher oxidation capability. Among several narrow band gap p-type materials, Ag_2_O has great potential in coupling with ZnO. Consequently, the visible-light harvesting ability could be improved by the formation of p-n heterojunction, as reported in the literature, as shown by Ag_2_O/ZnO/rGO [10], Ag/ZnO/CeO_2_ [11], and Ag_2_O/WO_3_ [12]. It is known that band gap, adsorption ability, charge carrier transfer, and separation efficiency are the key factors for enhancing the performance of the photocatalysts. Therefore, more recently, the application of plasmonic metals such as gold, silver, and copper has become a vital technology for improving the visible-light absorption by localized surface plasmonic resonance (LSPR), enhancing the charge transfer, and reducing the photogenerated charge carrier recombination rate through the formation of Schottky junctions with semiconductors [13,14]. Silver has been featured as an engineering photocatalyst over the other noble metals due to its unique advantages such as low cost, high work function, and oxygen adsorption ability. So far, several researchers have reported Ag-loaded photocatalysts, including Ag/BiOI/AgI [15], Ag/Ag_2_MoO_4_/ZnO [16], Ag/Bi_2_O_2_CO_3_/Bi_2_O_3_ [17], Ag/Zn_2_SnO_4_ [18], and Ag/TiO_2_/g-C_3_N_4_ [19]. Thus, metal-semiconductor heterojunction composite photocatalysts were extensively studied by exploiting the LSPR concept, and the conduction electrons of the metal nanoparticles then migrated to the conduction band of the semiconductor. 

In this work, Ag-Ag_2_O-ZnO nanocomposite was synthesized by a feasible one-step hydrothermal process. Methylene blue (MB), methyl orange (MO), and rhodamine B (RhB) dyes were selected as target pollutants for investigating the photocatalytic performance of the synthesized nanocomposite heterojunction photocatalyst. The nanocomposite remarkably enhanced the photocatalytic performance under visible light irradiation, and the enhanced photocatalytic activity was explored by investigation of the structure, morphology, chemical composition, and optical properties.

## 2. Materials and Methods

Pure ZnO and Ag-loaded ZnO composite powders were synthesized by a hydrothermal method using Zinc nitrate [Zn(NO_3_)_2_.H_2_O], Silver nitrate [AgNO_3_], NaOH, and solvents, which were purchased from Sigma-Aldrich. The chemicals used in this experiment were of analytical grade and were used directly without any further purification. In a typical synthesis procedure, 2 g of zinc nitrate and different weight percentages (0, 2, and 5%) of silver nitrate were dissolved in 120 mL of DI water and stirred for 40 min at room temperature using a magnetic stirrer. Then, 16 mL of 4 M NaOH solution was added dropwise to the above solution and sealed in a Teflon-lined stainless-steel autoclave system. The autoclave was ultrasonically treated for 30 min, maintained at 180 °C for 6 h, and then naturally cooled to room temperature. The solid produced by the hydrothermal treatment was collected by centrifugation at 12,000 rpm for 4 min, then subsequently washed with DI water and ethanol several times, and finally dried at 80 °C in an oven for 12 h.

The crystalline structure of the synthesized composite powders was investigated by X-ray diffraction (XRD; MiniFlex600, Rigaku, Tokyo, Japan) using Cu K_α_ radiation (λ = 1.5406 Å) with a 2θ scanning range of 20–80° and a scanning rate of 4°/min. The surface morphology of the powders was analyzed by field-emission scanning electron microscopy (FE-SEM; TESCAN-MIRA2, Brno, Czech Republic) operating at an accelerating voltage of 20 kV. The elemental concentration was determined by energy-dispersive X-ray spectroscopy (EDX). The optical absorption spectra were recorded in the wavelength range of 250–900 nm by a UV-vis-NIR spectrophotometer (UV-3600, Shimadzu, Kyoto, Japan). Fourier transform infrared (FT-IR) spectra were recorded in the range of 4000 to 500 cm^−1^ by a Spectrum 100, PerkinElmer FTIR spectrometer. The multi- and single-point Brunauer-Emmett-Teller (BET) surface areas of the samples were determined by using the Quantachrome QuadraSorb SI surface area analyzer instrument operating at 77.3 K accompanied by QuadraWin software. The samples were outgassed at 80 °C for 2 h before the surface area measurements. The chemical composition and bonding states of the photocatalysts were studied by X-ray photoelectron spectroscopy (XPS) by a Thermo Scientific K-Alpha spectrometer using an Al Kα X-ray source with a constant analyzer mode. Photoluminescence spectra of the samples were recorded using a Perkin Elmer LS-50B luminescence spectrometer.

The photocatalytic activity of pure ZnO and Ag/Ag_2_O/ZnO composites were evaluated for the degradation of three types of organic dyes: methylene blue (MB), methyl orange (MO), and Rhodamine B (RhB) under 300 W Xe-lamp equipped with a 420 nm cut-off filter. The photodegradation experiment was performed at ambient temperature, 100 mg photocatalyst was added to a photoreactor containing 100 mL of 10 ppm organic dye (MB, MO, and RhB). Before irradiation, the organic dye solution was stirred for 30 min in dark conditions to ensure adsorption–desorption equilibrium between the dye solution and the photocatalyst. The dye solution was then exposed to visible light while stirring. At given time intervals, aliquots were collected from the photoreactor and then centrifuged at 10,000 rpm for 2 min to separate the remnant photocatalyst from the dye solution, and then the concentration of dye solution in the supernatant was monitored, measuring the absorbance by using the UV-vis-NIR spectrophotometer. The dye degradation efficiency (*D*%) was calculated by the following equation: $D\%=\left(C_{0}-C/C_{0}\right)\times 100\%$, where *C*_0_ is the initial concentration and *C* is the degradation concentration of the dye under visible light irradiation. 

## 3. Results and Discussion

The crystal structures of ZnO and Ag-ZnO with different Ag contents were investigated by XRD, as shown in Figure 1a. The results demonstrated that peaks appeared at 31.88, 34.5, 36.36, 47.7, 56.76, 63.08, 66.55, 68.17, and 69.3° corresponding to the (100), (002), (101), (102), (110), (103), (200), (112), and (201) crystal planes of the hexagonal ZnO (JCPDS: 01-079-0205), respectively. The diffraction peaks observed at 38.2, 44.4, and 64.5° are assigned to the (111), (200), and (220) crystal planes of the cubic phase silver nanoparticles (JCPDS: 01-087-0597). The diffraction peak intensity of Ag increased with an increase in Ag concentration. In contrast, the 5% Ag-ZnO samples exhibited additional diffraction peaks at 32.9, 38.13, and 55.04, which can be indexed to the (111), (200), and (220) planes of the cubic Ag_2_O phase (JCPDS: 01-075-1532), indicating that a Ag_2_O/ZnO heterojunction was successfully formed in the 5% Ag-ZnO samples. The ZnO diffraction peaks in the Ag-loaded composite powders had no evident peak shift compared to the pure ZnO, which indicates that Ag was loaded on the surface of the ZnO but not incorporated into the ZnO lattice. No other additional impurity peaks were observed aside from the ZnO, Ag_2_O, and Ag, which indicates the phase purity of the synthesized materials. The average crystallite size of ZnO, Ag_2_O, and Ag of the 5% Ag-ZnO calculated by using Scherrer’s equation was about 34, 25, and 28 nm, respectively. A BET analysis is typically performed over the region (which is near the completed monolayer formation) of relative pressures ranged 0.05 < P/P_0_ < 0.3 to determine the specific surface area of the materials [20]. Figure 1b shows the N_2_ adsorption isotherms of pure ZnO, 2% Ag-ZnO, and 5% Ag-ZnO. The corresponding BET surface areas were determined to be 4.4 m^2^/g, 6.38 m^2^/g, and 9.4 m^2^/g, respectively. The Ag nanoparticle addition increased the surface area of the nanomaterials and the 5% Ag added ZnO powders showed the highest surface area.

Figure 2 displays the FE-SEM images of the pure ZnO and Ag-loaded ZnO composite powders. The pure ZnO shows self-assembled densely arranged anisotropic ZnO nanorods (length 1–2 μm) with a three-dimensional flower-like microstructure. The Ag-loaded ZnO shows a similar three-dimensional microstructure comprised of ZnO nanorods with randomly distributed spherical shape Ag nanoparticles. The element constituents of 5% Ag-ZnO were obtained by the EDX spectra shown in Figure 2d, and the table presented in the inset verifies the existence of Zn, O, and Ag elements of about 76.8, 18.0, and 5.2 wt.%, respectively. Furthermore, the elemental distribution of Zn, Ag and O elements was successfully confirmed by SEM-EDX elemental mapping, as shown in Appendix A. The internal microstructure of the 5% Ag-ZnO was further characterized by TEM, as shown in Figure 2e. A spherical shape Ag nanoparticle of a rough size of 90 nm was observed over the ZnO nanorod of a rough width of 140 nm. A high-resolution TEM image of the partial region encircled in Figure 2e at the interface of ZnO and Ag was further observed to explore the heterostructure, and as shown in Figure 2f, the lattice spacing of ZnO and Ag was 0.246 and 0.236 nm, corresponding to the (101) and (111) lattice planes, respectively. Moreover, the cross-lattice pattern observed at the ZnO/Ag interface corroborates the formation of the heterostructure, and the layered formation of overlapping lattice fringes at the edge of Ag could represent the Ag_2_O phase formation. These observations imply the surface distribution of Ag over the ZnO, which could offer a strong interfacial interaction that leads to enhanced charge carrier transfer.

XPS was performed to determine the chemical component and bonding configuration of the ZnO, 2% Ag-ZnO, and 5% Ag-ZnO. The full scan survey spectrum shown in Figure 3a revealed that there no other peaks belonging to any impurity elements other than the Zn, Ag, O, and C were observed, which indicates the high phase purity of the synthesized photocatalyst. The high-resolution XPS spectra of 5% Ag-ZnO of Zn 2p, Ag 3d, and O 1s are shown in Figure 3b–d, and high-resolution XPS spectra for pure ZnO and 2% Ag-ZnO are shown in Appendix A. The Zn 2p spectra deconvoluted into two peaks centered at binding energy 1021.4 and 1044.5 eV, with the binding energy splitting of 23 eV being ascribed to the spin-orbit of Zn 2p_3/2_ and Zn 2p_1/2_ components, respectively [21], which substantiated the existence of Zn in the form of Zn^2+^ of the ZnO phase. The Ag 3d peaks were deconvoluted four peaks as shown in Figure 3c. The two binding energy positions at 367.3 and 373.3 eV with a binding energy separation of 6 eV belonging to Ag 3d_5/2_ and Ag 3d_3/2_ of the metallic silver, respectively, are in agreement with the previous report [22]. The other two binding energy positions at 374.2 and 368.2 eV are attributed to Ag 3d_5/2_ and Ag 3d_3/2_ components of silver in the form of Ag–O, respectively, which are in agreement with the reported Ag_2_O phase [23,24]. In addition to the XRD results, the XPS studies proved that the 5% Ag-ZnO consists of Ag-Ag_2_O-ZnO heterostructures. Besides, Ag did not affect the Zn 2p peak position, which confirms that the Ag was loaded on ZnO nanorods. The two peaks of O 1s located at 530.1 and 531.6 eV correspond to the crystal lattice oxygen (Zn–O) and surface chemisorbed oxygen species, respectively. The functional groups that exist in the ZnO and Ag-ZnO samples were further confirmed by the FT-IR analysis shown in Appendix A. The broad absorption bands around 3374 cm^−1^ were assigned to the stretching vibrations of the hydroxyl (O-H) group, and 1640 cm^−1^ corresponded to the first overtone of the crucial stretching mode of OH due to adsorbed water molecules on the surface of ZnO [25]. The peaks observed at 870 cm^−1^ and 678 cm^−1^ were attributed to OH twisting vibrations and the Zn-O characteristic peak, and 512 cm^−1^ was designated to the Zn-O stretching vibration in the ZnO lattice [26].

As shown in Figure 4a, UV-vis-NIR absorption spectra were evaluated to determine the photo-absorption behavior of the prepared ZnO and Ag-ZnO composite powders. The absorption edge was redshifted towards the longer wavelengths with an increase of the Ag concentration, indicating the visible light response of the samples. Moreover, the absorption was significantly enhanced throughout the visible region (400–800 nm) due to the LSPR owing to Ag, which is advantageous for enhancing the activity of the photocatalyst. Optical band gap $\left(E_{g}\right)$ was determined by Tauc’s plot and the linear extrapolation of the (αhv)^2^ versus hv (Figure 4b), using the following equation:$$\alpha hv=C\;{\left(hv-E_{g}\right)}^{2}$$
where α is the absorption constant, *C* is a constant, and hv is the incident photon energy. The band gap of the 5% Ag-ZnO was decreased from 3.15 eV (pure ZnO) to 2.8 eV due to the addition of Ag and Ag_2_O phases. A similar reduction in the optical band gap of the ZnO heterojunction composites was reported by Xu et al. [10]. These results suggest an enhanced ability to capture the visible light, which facilitates a large amount of charge carrier generation, thereby improving the photocatalytic ability. 

Photogenerated charge carrier separation efficiencies were studied by Photoluminescence (PL) spectra. The PL intensity is directly proportional to the recombination rate of the charge carriers. As shown in Figure 5a, a significant decline in the PL intensity of the 2% Ag-ZnO and 5% Ag-ZnO samples was observed compared to the pure ZnO samples, indicating the superior separation rate of the photogenerated charge carriers, which are thereby conducive to enhanced photocatalytic performance. Time-resolved photoluminescence spectra (TRPL) were analyzed to further investigate the photogenerated charge transfer dynamics of the pure ZnO, 2% Ag-ZnO, and 5% Ag-ZnO samples, and the corresponding TRPL spectra were shown in Figure 5b. The decay curves were fitted using the bi-exponential decay model by the following equation: I=α exp(−tτ1)+β exp(−tτ2)
where *I* is the PL intensity, *t* is the time constant, *α* and *β* are PL amplitudes, and *τ*_1_ and *τ*_2_ are the radiative recombination and nonradiative relaxation processes of photoinduced electron-hole pairs, respectively [11]. The average charge carrier lifetime (*τ*_avg_) was determined using the following equation:
$$\tau_{avg}=\left(\alpha \tau_{1}{}^{2}+\beta \tau_{2}{}^{2}\right)/\left(\alpha \tau_{1}+\beta \tau_{2}\right)$$

The corresponding fitted parameters and the charge carrier lifetime were listed in Table 1. The lifetimes of the pure ZnO, 2% Ag-ZnO, and 5% Ag-ZnO samples were 5.23, 6.44, and 10.02 ns, respectively. The average lifetime for the 5% Ag-ZnO samples was greatly prolonged, which confirms that effective charge carrier separation was accomplished. The enhanced charge carrier separation efficiency and average lifetime in the 5% Ag-ZnO samples could be attributed to the formation of a ZnO/Ag_2_O heterojunction and the LSPR phenomena due to the presence of Ag. Thus, the activity of the Ag/Ag_2_O/ZnO photocatalyst obtained by 5% Ag-ZnO could be greatly improved by superior charge carrier separation efficiency.

Photocatalytic degradation tests for three types of organic dyes i.e., methylene blue (MB), methyl orange (MO), and rhodamine B (RhB) were performed under visible-light irradiation. Photocatalytic degradation with respective light illumination times are shown in Figure 6a–c for the 5% Ag-ZnO sample. The three types of organic dye degradation efficiency results of pure ZnO, 2% Ag-ZnO, and 5% Ag-ZnO with respect to visible light irradiation time are shown in Figure 6d–f. Obviously, the dye concentration was gradually decreased upon visible light illumination time. The 5% Ag-ZnO displayed the highest photocatalytic activity in 60 min with a degradation efficiency of 97.3, 91.1, and 94.8% for MB, MO, and RhB, respectively. Appendix A depicts the comparison of the photocatalytic removal efficiencies of the ZnO and Ag-ZnO samples for all three types of organic dyes. The photocatalytic activity kinetic reaction can be described by the pseudo-first-order kinetics:$$-ln\left(C/C_{0}\right)=kt$$
where *k* is a rate constant, and *t* is the light irradiation time. The slope of the fitted lines shown in Figure 6d–f indicates the first-order kinetic rate constant (*k*). The photocatalytic organic dye degradation efficiency and the calculated k value for the pure ZnO, 2% Ag-ZnO, and 5% Ag-ZnO are given in Appendix A for MB, MO, and RhB. The 5% Ag-ZnO showed the highest photocatalytic rate constant of 5.7 × 10^−2^, 3.7 × 10^−2^, and 5.1 × 10^−2^ min^−1^ for the MB, MO, and RhB, respectively. The stability and reusability of the 5% Ag-ZnO photocatalysts were studied to degrade the RhB dye in five repeated cycles, as shown in Figure 7b. The 5% Ag-ZnO photocatalyst showed stable dye-degradation performance over five repeated cycles, and the structure stability after repeating photocatalytic tests was further confirmed by XRD, as shown in Figure 7c. The significant enhancement in photocatalytic activity of the photocatalyst could be understood by the mechanism demonstrated in the schematic diagram Figure 7d. As shown in the schematic, only the Ag_2_O can be excited under visible light irradiation, the photogenerated electrons–holes electrons could be transferred from the CB of Ag_2_O to the CB of ZnO due to the strong interfacial contact between the catalysts. Thus, the photogenerated charge carriers could be effectively separated due to p-n heterojunction formation. At the heterojunction equilibrium, the Ag_2_O possesses a negative charge, and the ZnO possess a positive charge, generating an opposing electric field. The CB of ZnO (−0.85 eV) lies below the Ag_2_O (−1.3 eV), and the work function of ZnO (5.3 eV) is higher than that of Ag_2_O (4.6 eV). As reported by Kadam et al., charge transfer at the interface is more thermodynamically favorable due to the reduced existence of a barrier between the ZnO/Ag_2_O heterostructure [27]. Consequently, the electrons transfer from Ag_2_O to ZnO, while the holes can move to Ag_2_O. Due to this p-n heterojunction formation, separation of the photogenerated charge carriers is greatly enhanced, which can be confirmed by the PL spectra. As the CB potential of Ag_2_O was more negative than O_2_/.O_2_^−^ (−0.33 eV), superoxide radicals were generated from the reaction of dissolved oxygen molecules and electrons on the surface of the ZnO. Additionally, the holes in the VB of Ag_2_O react with OH to the hydroxyl radicals. The Ag can form the Schottky junction with ZnO, which acts as a sink for the electrons, thereby promoting the interfacial charge transfer kinetics between Ag/ZnO, leading to the separation of photoexcited charge carriers [28]. In addition, Ag has the ability to cause the surface plasmon-driven electrons to react with O_2_ to form .O_2_^−^ [29]. The photocatalytic performance of the synthesized metal-semiconductor heterojunction structure i.e., Ag/Ag_2_O/ZnO achieved by 5% Ag-ZnO was compared with the previously reported heterojunction-based photocatalysts as shown in Table 2. Obviously, in comparison, the study shows outstanding photodegradation performance. Thus, the photocatalytic performance of the Ag/Ag_2_O/ZnO metal-semiconductor heterojunction photocatalyst is greatly enhanced under visible light.

## 4. Conclusions

In summary, we demonstrated the utilization of surface plasmonic metal loaded p-n heterojunction Ag-Ag_2_O-ZnO nanocomposites synthesized by a hydrothermal method for the visible-light-driven photocatalytic degradation of organic dyes. The structural and chemical analysis results substantiated the Ag_2_O phase formation in 5% Ag-ZnO powders. The optical properties revealed that the photocatalyst showed enhanced absorption in the entire visible region, which confirms the strong interaction of plasmonic Ag nanoparticles with light. The optical band gap was shifted to the visible region from 3.1 eV to 2.8 eV owing to the heterostructure formation. The photogenerated charge carrier dynamics investigated by PL and TRPL showed that the photocatalyst exhibited significant charge carrier separation efficiency with a two-fold increment in the average lifetime. Three different organic dyes, MB, MO, and RhB were employed to study the photocatalytic performance. The Ag-Ag_2_O-ZnO photocatalyst achieved by 5% Ag-ZnO showed superior photocatalytic activity with significant stability and reusability tested for five repetitive cycles. Based on the results, it can be deduced that the photocatalytic activity accomplished by Ag-Ag_2_O-ZnO nanocomposite photocatalyst was enhanced, owing to the combination of LSPR and p-n heterojunction formation. Consequently, the photocatalyst has potential application for removing the undesirable organic pollutants in wastewater for environmental remediation. 

## Figures and Tables

**Figure 1 nanomaterials-12-02528-f001:**
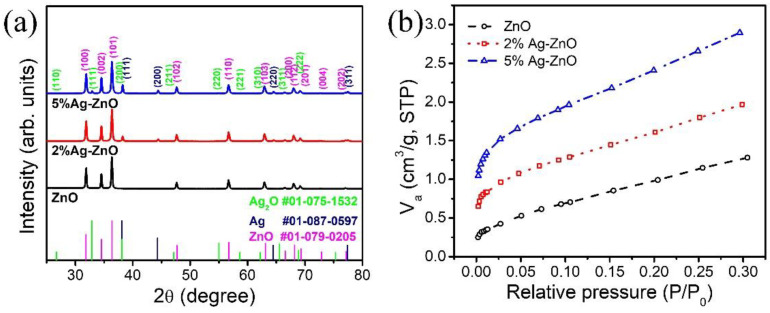
(**a**) X-ray diffraction pattern, and (**b**) N_2_ adsorption isotherms of the pure ZnO, 2% Ag-ZnO, and 5% Ag-ZnO powders.

**Figure 2 nanomaterials-12-02528-f002:**
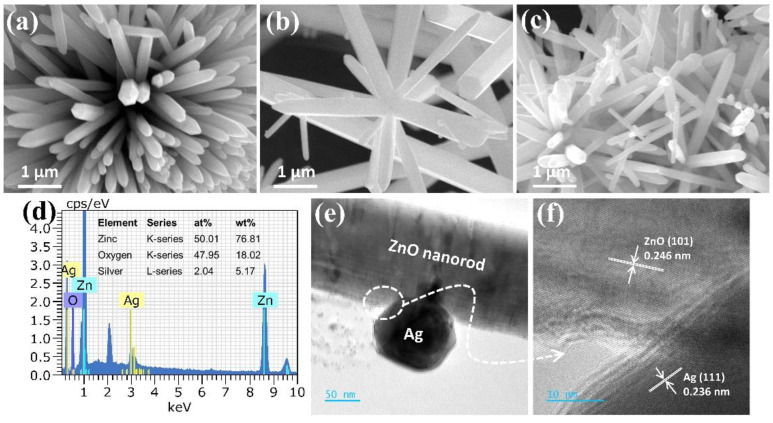
SEM image of pure ZnO (**a**), 2% Ag-ZnO (**b**), and 5% Ag-ZnO (**c**); EDX spectra of the 5% Ag-ZnO (**d**); low-magnification (**e**), and high-resolution (**f**) TEM images of 5% Ag-ZnO.

**Figure 3 nanomaterials-12-02528-f003:**
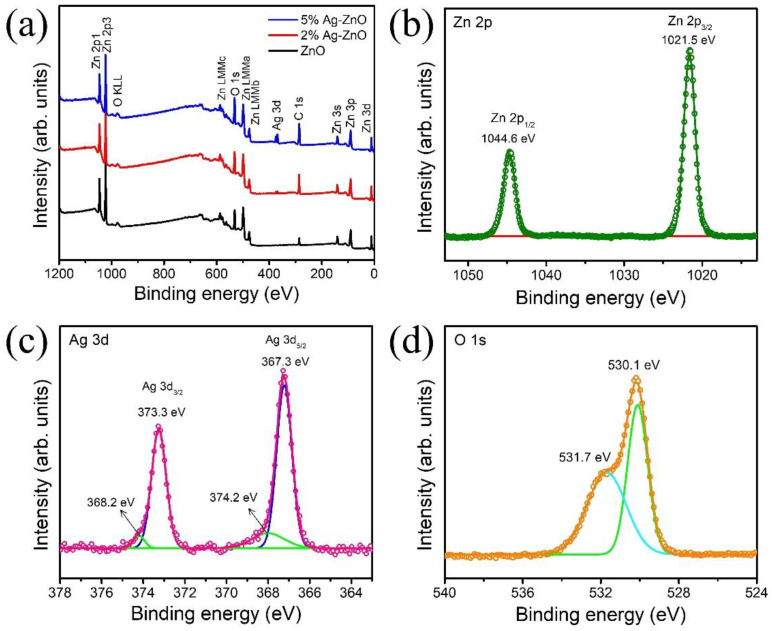
(**a**) Full scan survey spectra of pure ZnO and Ag-ZnO, and high-resolution XPS spectra of the Zn 2p (**b**), Ag 3d (**c**), and O1s (**d**) of the 5% Ag-ZnO.

**Figure 4 nanomaterials-12-02528-f004:**
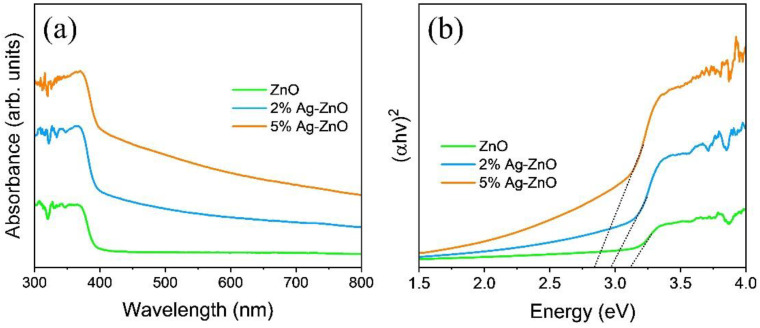
(**a**) UV-visible absorption spectra, and (**b**) Tauc plot of the pure ZnO and Ag-ZnO.

**Figure 5 nanomaterials-12-02528-f005:**
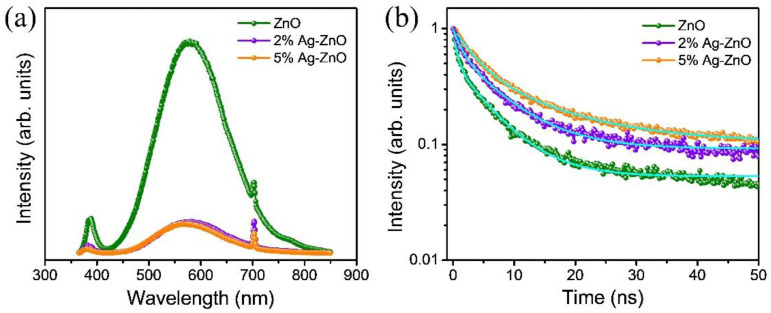
(**a**) Photoluminescence spectra, and (**b**) time-resolved photoluminescence spectra of the pure ZnO, 2% Ag-ZnO, and 5% Ag-ZnO.

**Figure 6 nanomaterials-12-02528-f006:**
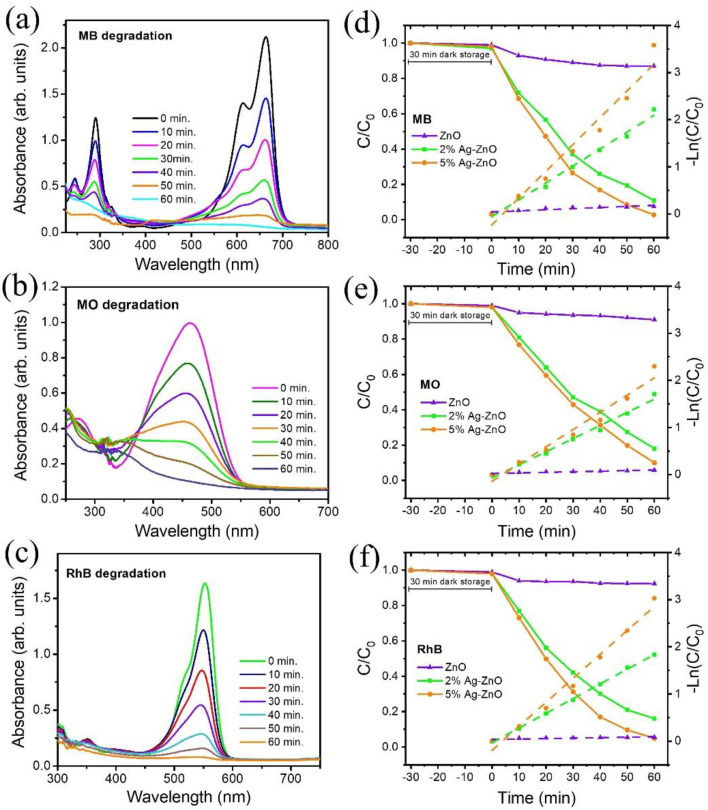
(**a**–**c**) Photocatalytic degradation performance of the 5% Ag-ZnO under visible light irradiation; (**d**–**f**) Photocatalytic degradation efficiency and pseudo-first-order kinetics of pure ZnO, 2% Ag-ZnO, and 5% Ag-ZnO for MB, MO, and RhB.

**Figure 7 nanomaterials-12-02528-f007:**
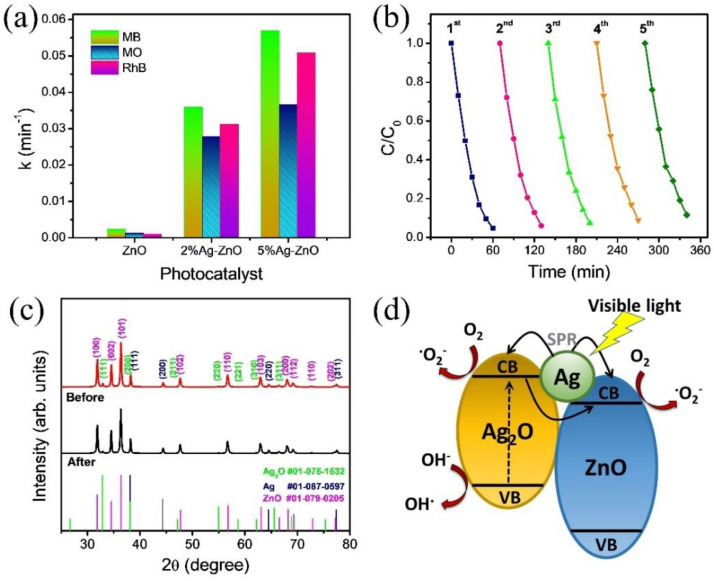
(**a**) Comparison of the pseudo-first-order kinetic rate constant of ZnO and Ag-ZnO, (**b**) cyclic stability test conducted for five repeated cycles of 5% Ag-ZnO sample for RhB; (**c**) X-ray diffraction pattern of 5% Ag-ZnO sample before and after the cyclic test for five cycles, and (**d**) schematic illustration of the possible mechanism for enhanced photocatalytic performance by heterojunction structure formation of Ag-Ag_2_O-ZnO.

**Table 1 nanomaterials-12-02528-t001:** The TRPL fitting parameters and the average lifetime of the pure ZnO, 2% Ag-ZnO, and 5% Ag-ZnO samples.

Sample	*τ*_1_ (ns)	*α*	*τ*_2_ (ns)	*β*	*τ*_avg_. (ns)
ZnO	0.61	0.48	5.75	0.45	5.23
2% Ag-ZnO	1.33	0.33	7.01	0.56	6.44
5% Ag-ZnO	12.7	0.36	3.43	0.54	10.02

**Table 2 nanomaterials-12-02528-t002:** Comparison between the photocatalytic performance of the prepared photocatalyst and some previous studies.

S.No	Photocatalyst	Light Source	Organic Pollutants and Concentration	Irradiation Time (min)	Degradation Efficiency	Rate Constant, k (×10^−2^ min^−1^)	Ref.
1	TiO_2_-ZnO	Xe lamp, simulated visible light	RhB, 10 mg/L	180	89%	1.1	[6]
2	Co_3_O_4_-ZnO	Solar light	MO, 10 mg/L	120	98%	-	[8]
3	Ag-ZnS_2_O_4_	Xe lamp, simulated solar light	MB, 10 mg/L	120	94%	2.3	[18]
4	Ag-AgBr-ZnO-rGO	105 W, Fluorescent lamp, visible light	MO, 10 mg/L	80	97.2%	4.6	[30]
5	Ag/ZnO-ZnFe_2_O_4_	200 W, Xe lamp, simulated sunlight	MB, 10 mg/L	100	93%	2.1	[31]
6	CuO-Cu_2_O	150 W, Xe lamp, visible light	MB, 5 mg/L MO, 5 mg/L	240	90% 60%	1.0 0.4	[32]
7	Au-ZnO	Visible light	RhB, 10 mg/L	250	97%	1.2	[33]
8	Ag_2_O-ZnO-PVDF	600 W, Xe lamp, visible light	MB, 10 mg/L	600	61%	0.68	[34]
9	Au-B-TiO_2_	300 W, Xe lamp, visible light	RhB, 10 mg/L	60	82%	2.8	[35]
10	Ag-Ag_2_O-ZnO	300 W Xe lamp, visible light	MB, 10 mg/L MO, 10 mg/L RhB, 10 mg/L	60	97.3 91.1 94.8	5.7 3.7 5.1	This study

## Data Availability

Not applicable.

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
