# Peer review of "Enhanced Visible-Light-Driven Photocatalysis of Ag/Ag2O/ZnO Nanocomposite Heterostructures"

_nanomaterials, 2022, doi:10.3390/nano12152528_

Round 1

Reviewer 1 Report

This manuscript reported the preparation of Ag/Ag2O/ZnO heterostructures, which can be used for the photocatalytic degradation of photocatalyst tested for methylene blue, methyl orange, and rhodamine B. This Ag/Ag2O/ZnO heterostructure exhibited a remarkable value for the degradation capacity. My recommendation is to reconsider after the Major revisions.

My comments:

1.        Please do not use "excellent" in the title, the author should revise it.

2.        Abstract: In the abstract, the results of the research must be briefly described. Opening sentences should be carried over to the introduction section.

3.        Abstract requires more technical achievements from the proposed work to highlight the novelty of the work.

4.        Keywords should not identical to title words. Should revise it.

5.        The authors should clearly explain the innovation, research gap, market gap, market demand, and importance of their work in the manuscript's introduction. They should justify the value of the work and compare their work with previously similar published papers. They should develop the electrocatalysis advantage and applications compared to other known systems. The introduction section needs to be elaborated.

6.        Figure 5b, X-axis unit having typo mistake “(ns)”. The author should rectify this.

7.        Line No. 281, 282, the authors should reverify the schematic mechanism diagram, as seen in Fig. 7d, by providing an appropriate reference.

8.        In the comparison to Table (Page No. 10), the author should include recent and relevant references to understand the efficiency of the proposed work. Also, the author should compare the proposed work with other catalysts than ZnO-based catalysts.

9.        Author must give UV-vis reflection spectra of each sample.

10.     Fig. S2, FT-IR spectra, authors must explain all the peaks? Several supplementary peaks have not been assigned with any information and major peaks should mark in the Figure for the reader's better understanding.

11.     In FTIR spectra of when several spectrums merged in one plot, Y-axis should give as “Transmittance (a.u.)” not in (%). Should revise it.

12.     Removal percentage and qe(mg/g), which to depend on to qualify adsorbent quality? Authors should plot both qe(mg/g) and removal(%) in Figure S3.

13.     It is required that the author demonstrate the flaws and potential improvements in the conclusion.

14.     The grammar of the manuscript should be polished; the typos and grammatical errors are scattered throughout the paper and need to be corrected with utmost care.

Author Response

We express our sincere thanks to the reviewer for taking the necessary time and effort to review our manuscript. We sincerely appreciate all your valuable comments and suggestions, which helped us in improving the quality of the manuscript. Detailed response to each comment was given in a separate file. Thank you. 

Sincerely,

corresponding author: Prof. Kee-Sun Lee

Reviewer 2 Report

In this article, the authors investigated the improved photocatalytic activity of ZnO-Ag nanostructures. In brief, the article can be reconsidered for publication after a revision. 

1) Provide EDX elemental mapping to confirm the uniform distribution of Ag on the ZnO surface. 

2) Ag concentration (% wt.) should be determined quantitatively by ICP-MS, ICP-OES, or AAS method because part of Ag can be lost during synthesis and removed with a solution. 

3) What was the adsorption of dyes on ZnO-Ag after 30 min in the dark? Basically, C/Co in Figures 6d-f should not be started from 1, it should be less due to adsorption. 

4) Statistical information is not shown, how many samples were tested per trial, and where are the error bars for each measured point in Figures 6d-f? 

5) Although the PL is a good indicator for improved charge-separation, it is recommended to perform experiments with radical scavengers, it will show the mechanism of photocatalytic improvement. 

6) Introduction part can be improved, i.e. photocatalytic activity can be improved with the help of optical materials (doi: 10.1007/s11706-019-0482-z) and 2D materials (doi: 10.1039/C6NR00546B).    

Author Response

We express our sincere thanks to the reviewer for taking the necessary time and effort to review our manuscript. We sincerely appreciate all your valuable comments and suggestions, which helped us in improving the quality of the manuscript. Detailed response to each comment was given in a separate file. Thank you.

Sincerely,

correspondig author: Prof. Kee-Sun Lee

Round 2

Reviewer 1 Report

Author improved manuscript quality in the revision stage. The authors incorporated all requirements raised by the reviewer. Now it can be accepted for publication as it is. 

Reviewer 2 Report

A revised manuscript can be accepted for publication!